# Mechanical and Contractile Properties of Knee Joint Muscles after Sports-Related Concussions in Women Footballers

**DOI:** 10.3390/jfmk9020065

**Published:** 2024-04-07

**Authors:** Georgios Kakavas, Athanasios Tsiokanos, Michael Potoupnis, Panagiotis V. Tsaklis

**Affiliations:** 1Fysiotek Spine and Sports Lab, 11635 Athens, Greece; georgios.kakavas@gmail.com; 2ErgoMechLab, Department of Physical Education and Sport Science, University of Thessaly, 42100 Trikala, Greece; atsiokan@uth.gr; 3Medical School, 3rd Academic Orthopedic Clinic, Aristotle University of Thessaloniki, 54124 Thessaloniki, Greece; mikepot@otenet.gr; 4Department of Molecular Medicine and Surgery, Growth and Metabolism, Karolinska Institute, 17164 Solna, Sweden

**Keywords:** tensiomyography, knee, concussion, soccer, neuromuscular control

## Abstract

The purpose of this study was to determine if women footballers have an increased lack of neuromuscular control of the knee joint after a concussion compared to a healthy cohort tested with tensiomyography (TMG). Forty-one female collegiate footballers were enrolled in this study from which there were 20 with a history of sports-related concussions (SRCs) and 21 control subjects. Results from the SRC group had significantly higher Tc (ms) (z = −5.478, *p* = 0.000) and significantly lower Dm (mm) (z = −3.835, *p* = 0.000) than the control group in the case of the rectus femoris muscle. The SRC group had significantly higher Tc (ms) (z = −2.348, *p* = 0.016) and significantly lower Dm (mm) (z = −4.776, *p* = 0.000) than the control group in the case of the vastus medialis muscle. The SRC group had significantly higher Tc (ms) (z = −5.400, *p* = 0.000) and significantly lower Dm (mm) (z = −4.971, *p* = 0.000) than the control group in the case of the vastus lateralis muscle. The SRC group had significantly higher Tc (ms) (z = −5.349, *p* = 0.000) than the control group in the case of the biceps femoris muscle response, whereas no significant difference was found in Dm (mm) (z = −0.198, *p* = 0.853) between the groups. The results of the current study may have implications for current practice standards regarding the evaluation and management of concussions and can add valuable information for knee prevention programs as well.

## 1. Introduction

There are concerns present about the potential adverse effects of purposeful heading on cognitive function, postural control, and symptoms [1]. It has been suggested that there be mandatory bans on heading to prevent its potential acute and chronic effects [2]. Recently, heading was reported to cause 25–30% of concussions in football, and 62–78% of the concussions were from incidental head-to-head contact [3]. The term “concussion” comes from the Latin concussio, meaning “to strike together”.

Increased popularity and accessibility for female athletes has led to an increased incidence of sports-related injuries. Concussions make up a significant proportion of sports injuries and are associated with immediate and long-term consequences for youth athletes [4]. However, despite the increasing participation in female contact sports, there is relatively little research on gender-specific characteristics in concussions and their consequences. The most commonly referenced definition describes concussions as a brain injury induced by direct or indirect biomechanical force transmitted to the head, resulting in a reversible clinical syndrome manifested as signs and symptoms affecting the physical, cognitive, emotional, and sleep domains reflecting a predominantly functional, rather than structural, injury.

Female soccer players are at an increased risk of concussion when compared to their male counterparts [5]. While there is consensus in the sports medicine literature about the increased incidence, there is a lack of evidence as to why this is the case. The sex disparity in concussion rates in soccer players is a cause for concern with the growing population of females playing soccer and the evidence suggesting poorer outcomes following a concussion in the female sex [6]. A sports-related concussion (SRC) has been defined as a traumatic brain injury induced by mechanical forces resulting in the onset of temporary impairments of neurological function. While many SRCs follow a gradual recovery without long-term effects (vision problems, fatigue, headaches, and more), some cases can be prolonged and complex [7] (depression, anxiety, insomnia, and aggression). The pathophysiology of the neuroinflammatory response following SRCs has been hypothesized to correlate with concussion symptomatology and symptom duration. Interestingly, mild systemic inflammation seems to influence the SRC recovery process. Subjects with initial post-injury elevations in high-sensitivity C-reactive protein (hsCRP), an inflammatory biomarker, were more likely to experience persistent post-concussive symptoms, cognitive impairment, and ongoing psychological issues 3 months after the SRC [8]. There is a growing body of evidence that post-concussion neuromuscular control impairments are present during simple gait-related tasks such as obstacle navigation, obstacle clearance, gait initiation, and gait termination. 

Given the limited challenge posed by these gait tasks, neuromuscular control impairments are likely to be accentuated during high-demand athletic tasks [9]. Within the structure of the skeletal muscles, there are fascicles of muscular fibers that are made up of serially distributed contractile elements. These elements are controlled by the nervous system, control which results in obtaining the muscular strength required for movement and its control. According to many study designs, athletes appear to have an increased risk of sustaining a musculoskeletal injury following an SRC. Furthermore, dual-task neuromuscular control deficits may continue to exist after athletes report resolution of concussion symptoms or perform normally on other clinical SRC tests. Therefore, musculoskeletal injury risk appears to increase following an SRC and persistent motor system and attentional deficits also seem to exist after a concussion.

Tensiomyography used in this study measures the radial displacement of a muscle during an electrically evoked twitch contraction. The rate of muscle displacement is increasingly reported to assess contractile properties [10,11]. A muscle twitch is the contractile response to a single electrochemical signal of the nervous system or artificial electrical stimulation of the muscle. As such, a twitch provides information on muscle contractile properties and the functioning of the excitation–contraction coupling process. Tensiomyography [12] measures the radial displacement of the muscle belly during an electrically stimulated isometric twitch response. From the radial displacement curve, spatial and temporal parameters are derived. The two most frequently reported parameters are the maximum displacement (Dm) and the contraction time (Tc). Dm provides information on skeletal muscle stiffness and morphological and structural changes. Tc refers to the time interval between 10% and 90% of Dm and is correlated to the proportion of slow-twitch fibers. Therefore, a shorter Tc is commonly associated with a higher contraction velocity [13,14].

Our hypothesis is that the SRC group will have contractile alterations of the knee joint flexor and extensor muscles compared to a healthy cohort, with a focus on neuromuscular assessment using the tensiomyography method (TMG). These neuromuscular control impairments extend well beyond symptomatic recovery and fulfillment of return-to-play criteria. TMG parameters are crucial to understand the lack of neuromuscular control of the joint in healthy and injured athletes.

## 2. Methods and Materials

### 2.1. Subjects

Forty-one female collegiate footballers were enrolled in this study from which there were 20 with a history of SRCs and 21 control subjects. The athletes were recruited from the Greek Women Super League teams using the Head Count 2 week recall questionnaire [15]. (The Head Count 2 week questionnaire is a recall questionnaire regarding ball heading exposure in the duration of 2 weeks). The subjects recruited in both groups were in the same age range (18–22 years old). Specifically, the mean age of the SRC group was 19.35 ± 1.18 years and 20.10 ± 1.18 years in the control group, respectively (Table 1). Additionally, BMI was not statistically significantly different between the experimental (SRC) and control group (U = 175.500, *p* = 0.369 > 0.05). 

This study was approved by the Institutional Review and Bioethics Board, of The University of Athens, Department of Physical Education and Sports Science. (Application number 1453, approved 1 November 2023).The single testing TMG procedure was explained to all subjects, and they were asked to provide an informed consent before the TMG measurements took place. 

The inclusion criteria were that the SRC group must have a positive SCAT 5 TEST.

The exclusion criteria were that all subjects must not be injured in the lower leg the last 2 months prior to testing.

### 2.2. Instrumentation and Procedures

The TMG measurements were performed by a health professional who was not informed about the subjects’ SRC history. TMG works by using a device called myotonometer to measure the electrical activity of the muscles. The device sends a small electrical impulse into the muscle, causing the muscle fibers to contract. The myotonometer then measures the rate of contraction and degree of stiffness of the muscle, providing valuable information about the functional state of the tissue. The above device used is manufactured by TMG™, s2 model, and is made in Slovenia, EU. This is a noninvasive technique that reliably evaluates metrics including pain, contractility, and muscle function. TMG signals are analyzed in order to determine the following parameters: delayed time, contraction time, sustain time, relaxation time, and maximum amplitude. In our study, we two of them (delayed and contraction time).

### 2.3. Statistical Analysis

Data processing and statistical tests were carried out with SPSS 25. Initially, all the main data and characteristics of the sample of athletes were analyzed with descriptive statistics (central tendency, variance, distribution, and ranking indices). 

Testing for normality of the sample distribution was performed using the Shapiro–Wilk statistical criteria with a confidence interval set at 95%. Testing for normality of distribution was performed before each inferential statistical test of comparisons between the SRC and control groups. In the majority of cases, the results showed a non-normal distribution, and, therefore, non-parametric statistical tests were performed. Specifically, the Mann–Whitney statistical test was used to compare continuous variables between 2 independent groups (SRC and control groups). In all statistical tests, a 95% confidence interval (α = 0.05) was set, and the significance probability values were calculated using the Monte Carlo simulation method.

## 3. Results

### 3.1. TMG Parameters

The descriptive analysis of the TMG parameters of the muscular response for both the SRC and control groups are presented in Table 2. 

### 3.2. Mann–Whitney U Test

The results indicated the following:

-The SRC group had significantly higher Tc (ms) (z = −5.478, *p* = 0.000) and significantly lower Dm (mm) (z = −3.835, *p* = 0.000) than the control group in the case of rectus femoris muscle response (Figure 1).-The SRC group had significantly higher Tc (ms) (z = −2.348, *p* = 0.016) and significantly lower Dm (mm) (z = −4.776, *p* = 0.000) than the control group in the case of vastus medialis muscle response (Figure 2)-The SRC group had significantly higher Tc (ms) (z = −5.400, *p* = 0.000) and significantly lower Dm (mm) (z = −4.971, *p* = 0.000) than the control group in the case of vastus lateralis muscle response (Figure 3). -The SRC group had significantly higher Tc (ms) (z = −5.349, *p* = 0.000) than the control group in the case of biceps femoris muscle response, whereas no significant difference was found in Dm (mm) (z = −0.198, *p* = 0.853) between the groups (Figure 4). 

**Figure 1 jfmk-09-00065-f001:**
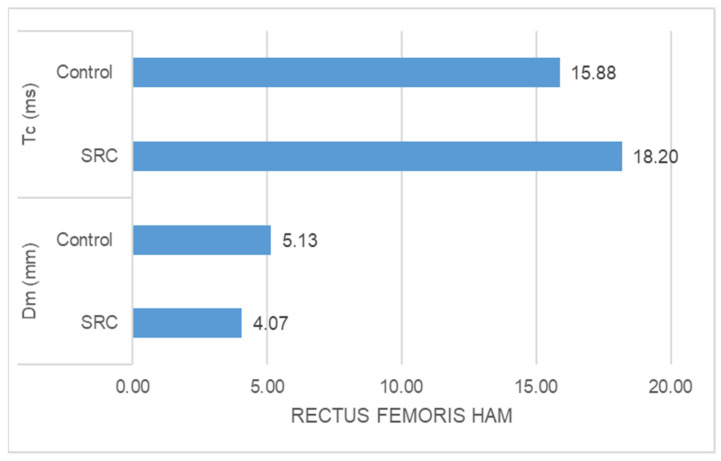
Rectus femoris response.

**Figure 2 jfmk-09-00065-f002:**
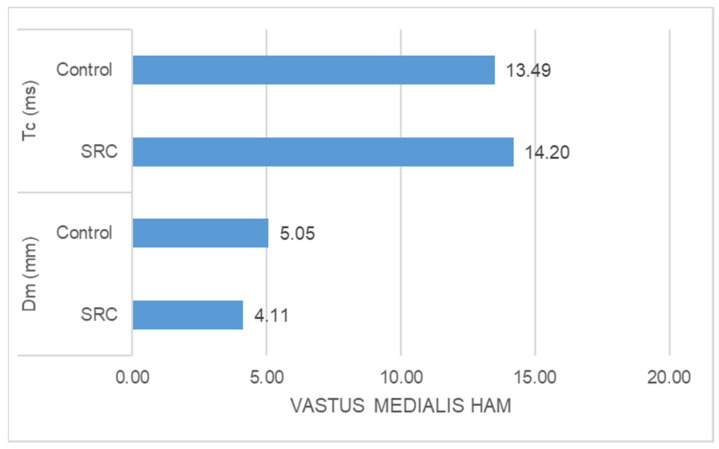
Vastus medialis response.

**Figure 3 jfmk-09-00065-f003:**
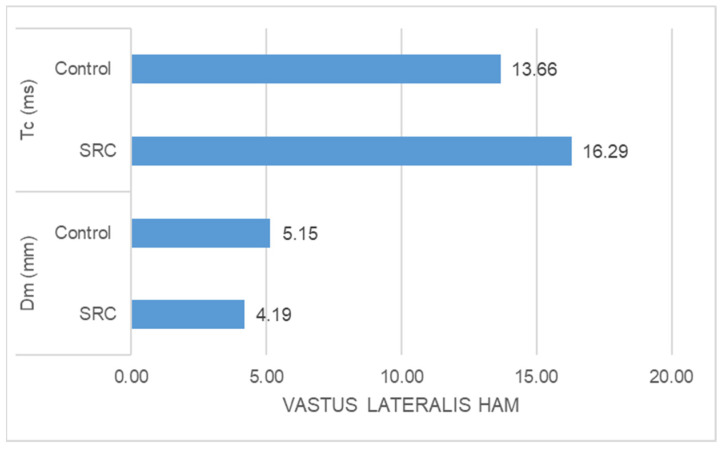
Vastus lateralis response.

**Figure 4 jfmk-09-00065-f004:**
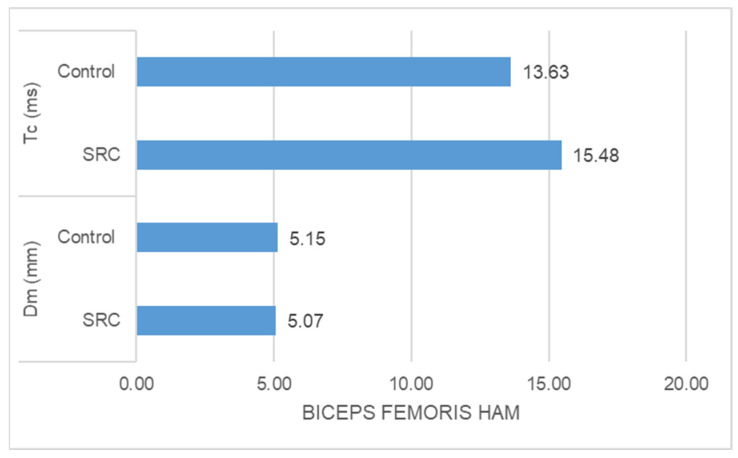
Biceps femoris response.

Table 3 shows the results of the Mann–Whitney U Test performed in the case of all four muscles assessed.

## 4. Discussion

Our hypothesis was that SRC group would have contractile alterations of the knee joint flexor and extensor muscles compared to a healthy cohort. The results of our paper confirms that, after a concussion, the muscles that control and move the knee joint are compromised. Our overall findings suggest that repetitive ball heading and SRCs in women footballers may be associated with biomechanically abnormal knee patterns in the extensor muscles of the knee compared to the flexor muscles, and this may increase the risk for future knee injuries in women footballers with a history of SRCs [16,17]. Repetitive head impacts are common in football due to the sport’s unique feature of purposeful heading of the ball with the unprotected head. This element of the game remains controversial, as the potential associations with long-term neurological consequences are still not settled [18]. Nevertheless, even though heading is normally asymptomatic and rarely causes concussion, there is evidence to suggest that cumulative heading exposure may lead to brain alterations in adults and affect cognitive function in adolescents.

The sensorimotor system encompasses all the afferent, efferent, and central integration-processing components involved in maintaining functional joint stability during body movements [19,20,21]. Inadequate functioning and the loss of neuromuscular control of this system may predispose toward joint trauma, a situation that commonly occurs after ligament injury. Poor neuromuscular control has been demonstrated to be a risk factor for musculoskeletal injury. An emerging area of research has identified that an increased risk of musculoskeletal injury may exist upon returning to sports after a sport-related concussion [22,23]. 

The mechanisms underlying this recently discovered phenomenon, however, remain unknown. One theorized reason for this increased injury risk includes residual neuromuscular control deficits that remain impaired despite clinical recovery. For example, abnormal movement patterns in the lower extremity and trunk have been shown to be different between high- and low-risk groups for anterior cruciate ligament injury; in particular, excessive frontal plane motion at the knee and poor trunk control have been found to be predictive of injury [24,25]. 

The potential for such an increased risk of musculoskeletal injury after a concussion has been observed in recently published studies on professional athletes. Nordstrom et. al. examined an injury registry comprised of 46 elite male professional soccer clubs from the February 2001 to December 2011 seasons [26,27]. Sixty-six cases of concussion were examined and were found to have a hazard ratio of 1.70 for sustaining an acute-onset musculoskeletal injury in the year after a concussion compared to the year prior to a concussion. The hazard ratios for acute-onset injury in comparison to non-concussed players ranged from 1.76 to 3.69, depending on the duration of time from the concussion injury [28,29].

Cross et. al. noted similar findings among professional rugby players, with the incidence of any injury for players who returned to play in the same season following a diagnosed concussion being 60% higher compared to non-concussed players. Finally, Pietrosimone et. al. employed a survey of 2429 retired professional football players and demonstrated an association of multiple injury types in players who also reported a history of concussion [30,31]. The number of injuries experienced was also found to increase with an increasing number of concussions. 

However, there are limitations to these data. In particular, some of the above studies lack controls for relative amounts of participation; hence, it is possible that the relationship between concussions and musculoskeletal injuries may have been a function of higher rates of exposure (e.g., starters versus reserve players).Importantly, little is known about which factors influence head impact exposure in youth football. Based on a study on female youth players, Harriss et al. found that the number of purposeful headers increased with age [32,33]. Chrisman et al. suggested that also sex may play a significant role [34]. Both studies, however, were careful to emphasize the limited generalizability of their conclusions, mainly due to smaller sample sizes and homogenous populations. In addition, these studies were conducted in North America, and potential differences in playing styles between countries might further limit the external validity of their findings [35].

Providing more accurate data on heading exposure and how this may be influenced by age and sex is key to assess risk. Musculoskeletal injury risk may need to be an item of consideration in the development of future guidelines and rehabilitation strategies. Current clinical tools may also be insensitive at detecting meaningful post-concussion deficits and may provide insufficient coverage of possible domains of effect (such as neuromuscular function). The continued development of clinical tests, potentially including novel testing paradigms incorporating measures of neuromuscular control, may be required to adequately assess athletes for suitability for return to play after an SRC. Finally, post-concussion rehabilitation strategies may also need to be altered to account for this relationship. This may include incorporating aspects of neuromuscular-based injury prevention strategies that have been used with successful evidence for reducing the risk of injuries such ankle sprains and ACL ruptures [36]. 

### 4.1. Limitations of the Study

This current study is not without its limitations. The data were collected at a single institution, which may limit the overall generalizability of the results. This study utilized historical data in a prospective design with a relatively small sample size. As such, comparisons of subsequent injury risk based on injury type or comparisons by sport or position were not feasible. Any effects of a history of prior lower extremity injury were not included in the current analysis. This decision was made due to multiple factors. One, we had a lack of reliable information on pre-injury history. There was also a lack of consensus on what would qualify as a meaningful prior injury history both in terms of the injury experienced and the historical timeframe for inclusion, given our limited sample size and the breadth of reported injuries.

### 4.2. Conclusions

The results of the current study may have implications for current practice standards regarding the evaluation and management of concussions. These neurocognitive and neuromuscular deficits are likely subtle and not easily detected with current assessment strategies, yet they may have a significant clinical impact on subsequent injury risk as such deficits are likely to be magnified with more challenging athletic tasks. Hence, these deficits may create a “window of susceptibility” to musculoskeletal injury following a return to play after a concussion. The duration of this “window” of musculoskeletal injury is not well understood. The presence of these findings following concussions implies that the current return-to-play guidelines may not be sufficient, as presently designed, to protect athletes from potentially significant post-concussion sequelae according to the FIFA, IOC, and other concussion protocol guidelines.

## Figures and Tables

**Table 1 jfmk-09-00065-t001:** Sample characteristics.

	Sample		SRC Group		Control Group	
	Mean	SD	Mean	SD	Mean	SD
*n*	(*n* = 41)		*(n* = 20)		(*n* = 21)	
Age	19.73	1.23	19.35	1.18	20.10	1.18
Weight (Kg)	61.95	7.62	59.15	8.22	64.62	6.05
Height (cm)	166.76	7.54	163.45	7.44	169.90	6.32
BMI	22.21	1.67	22.07	2.03	22.35	1.27

Legend: *n*, number; SD, standard deviation; BMI, body mass index.

**Table 2 jfmk-09-00065-t002:** TMG descriptive parameters of muscles’ response in groups.

				Values			
Muscle	Variable	N	Group	Mean	SD	Median	Mean Rank
Rectus Femoris	Tc (ms)	20	SRC	18.20	0.09	18.21	31.50
		21	Control	15.88	0.61	15.88	11.00
	Dm (mm)	20	SRC	4.07	0.78	4.10	13.65
		21	Control	5.13	0.77	5.10	28.00
Vastus Medialis	Tc (ms)	20	SRC	14.20	0.06	14.20	25.50
		21	Control	13.49	0.76	13.34	16.71
	Dm (mm)	20	SRC	4.11	0.45	4.10	11.85
		21	Control	5.05	0.38	4.99	29.71
Vastus Lateralis	Tc (ms)	20	SRC	16.29	0.58	16.31	31.35
		21	Control	13.66	0.87	13.70	11.14
	Dm (mm)	20	SRC	4.19	0.39	4.12	11.48
		21	Control	5.15	0.42	5.19	30.07
Biceps Femoris	Tc (ms)	20	SRC	15.48	0.32	15.55	31.25
		21	Control	13.63	0.75	13.76	11.24
	Dm (mm)	20	SRC	5.07	0.15	5.17	21.38
		21	Control	5.15	0.51	5.15	20.64

Legend: N, number; SD, standard deviation; Tc, contraction time; Dm, maximal displacement.

**Table 3 jfmk-09-00065-t003:** Mann–Whitney U Test analysis values between groups.

Muscle	Variable	z	*p* Value
Rectus Femoris Ham	Tc (ms)	−5.478	0.000
	Dm (mm)	−3.835	0.000
Vastus Medialis Ham	Tc (ms)	−2.348	0.016
	Dm (mm)	−4.776	0.000
Vastus Lateralis Ham	Tc (ms)	−5.400	0.000
	Dm (mm)	−4.971	0.000 ^b^
Biceps Femoris Ham	Tc (ms)	−5.349	0.00
	Dm (mm)	−0.198	0.853

Legend: Tc, contraction time; Dm, maximal displacement.

## Data Availability

Data is unavailable on-line, due to privacy or ethical restrictions. Data can be available, upon request to 1st author G.K.

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
