# Peer review of "Mechanical and Contractile Properties of Knee Joint Muscles after Sports-Related Concussions in Women Footballers"

_jfmk, 2024, doi:10.3390/jfmk9020065_

Round 1

Reviewer 1 Report

Comments and Suggestions for Authors

Word Consistency: Title says Women and manuscript says Female

Lines 52-54     Please add a citation for this comment since there is “evidence”; “There is a growing body of evidence that post-concussion neuromuscular control impairments are present during simple gait-related tasks such as obstacle navigation, obstacle clearance, gait initiation, and gait termination.”

Line 56            State “…contractile alterations in 3 of the quadriceps and the biceps femoris-long head muscles compared to a healthy cohort.” Or “contractile alterations in muscles that control and move the knee joint compared to a healthy cohort.” The muscles are above the joint and the study was restricted to these 4 only.

Line 160          Can the authors quantify “repetitive ball heading” to provide context, similar to a pitch count in baseball or softball.

Lines 159 -162            Edit sentence to the following if this is the intention of the authors’ point being made. “Our overall findings suggest that repetitive ball heading and SRC in women footballers may be associated with abnormal biomechanical knee patterns that may increase the risk for future LE injuries in women footballers with a history of SRC.”

Lines 172 -174            Please add a statement on why “Inadequate functioning” increases chance of joint damage, moreover what type of damage such as bone degeneration would provide context. Currently reads that this is a long-term concern not an acute injury.

Lines 241 -244            The sentence needs to be rewritten for improved flow when reading. Additionally, what are the authors basing the current RTP guidelines for concussion on, such as the National Football League, Premier League, National Athletic Trainers Association, etc.?

Author Response

Lines 52-54, reference 8

Line 56, word around replaced with above knee joint

Line 160, every player, according to her position does multiple ball headings in every game. We can not provide this stat.

Lines 159-162, edited as proposed

Lines 172-174, statement added

Lines 241-244, statement rephrased and guidelines addedd

Reviewer 2 Report

Comments and Suggestions for Authors

The study aims to investigate the impact of concussions on neuromuscular control of the knee joint in women footballers compared to a healthy control group tested with Tensiomyography (TMG). Among 41 participants (20 with sports-related concussions (SRC) history and 21 controls), results showed that the SRC group exhibited significant differences in contraction time (Tc) and displacement (Dm) compared to the control group: specifically, the Rectus Femoris, Vastus Medialis, and Vastus Lateralis of the SRC group had higher Tc and lower Dm. Biceps Femoris in the SRC group had Higher Tc but no significant Dm difference. The findings suggest that concussions may contribute to increased neuromuscular control deficits in knee joint muscles.

In general, please check throughout the whole manuscript since several parts did not match the recommendations of the journal, e.g., the number of words in the abstract should not exceed 200 words, font size, space before and after symbols (e.g., . or ,), and reference form. For other suggestions please see the texts below.

Abstract

1.       Please check the word numbers that should be fitted to the journal guideline.

2.       While the abbreviation "TMG" is explained in the passage, it's generally good practice to introduce the full term before using the abbreviation consistently.

Introduction

1.       Introduce "TMG" and "SRC" in full before using abbreviations consistently.

2.       The second paragraph is lengthy, please consider creating two more paragraphs for sports-related concussion (SRC) and tensiomyography (TMG).

3.       Please add a reference to the sentences in lines 48-50.

4.       Please give examples or details of the long-term effects caused by SRC.

5.       Please explain the long-term effects of SRC in the text.

6.       Please explain more or add more details about the contractile alterations linked to neuromuscular control impairments, especially the muscles around the knee joint.

7.       The abbreviation TMG should be located after the first presence of tensiomyography in line 61.

8.       Please add one more paragraph to summarize the objectives and hypotheses of the current study.

Materials and methods

1.       In 2.1. Subjects, please briefly explain about HEAD COUNT 2 weeks recall used in the study.

2.       In 2.1. Subjects, please add the approval number and date of the ethics in the text.

3.       Information of participants who volunteered in the study shown in the third paragraph should be moved to merge with the first paragraph since they share the same main idea of the paragraph (participants).

4.       Is there a statistical test used to test the differences in the demographic data? Please clarify and perform statistical tests to ensure that both groups are similar.

5.       In 2.2. Instrumentation and Procedures, please add more details myotonometer (e.g., brand, version, company, country, etc) and how it works, as well as the parameters provided by the device.

6.       Please explain more about the rate of contraction and degree of stiffness of the muscle derived from the myotonometer, and how to interpret the results.

7.       The section 2.3. Inclusion and Exclusion Criteria should be moved into section 2.1 participants.

8.       In section 2.4, only the Shapiro-Wilk test is enough for a small sample size and add the effect size.

Results

1.       Please change the subheading of the Results (e.g., section 3.2) and the caption of the table (e.g., Table 3) to make them align with the results or objectives of the study.  

2.       Please use the term Figure instead of Graph.

3.       Please add the significant symbol to the figure.

4.       Please add the details of the effect size in the text.

Discussion

1.       Please start the discussion with a summary of the objectives and the findings of the study.

2.       There was no interpretation or discussion about the findings according to the objectives of the study.

3.       Each paragraph of the discussion is lengthy; please reconstruct it to make it easy to follow. The passage covers a range of topics, but it could benefit from clearer organization and separation of ideas into distinct sections or paragraphs. This would improve readability and help the reader follow the logical flow of information.

4.       The passage repeats the point about the controversial nature of heading in football and its potential long-term neurological consequences. While this is an important point, it is mentioned multiple times, and some condensation or rephrasing may improve clarity.

Conclusion

1.       The conclusion is lengthy. Some points can be added in the discussion part.

2.       The passage presents a complex set of ideas. Consider breaking down the information into smaller paragraphs or sections to enhance clarity and readability. Each idea could be presented more cohesively for a smoother flow.

3.       The term "window of susceptibility" is mentioned multiple times. While it is a key concept, consider rephrasing or introducing synonyms to avoid redundancy.

Author Response

Abstract corrected as recommended

Introduction

  1. Introduced in abstract
  2. corrected as suggested
  3. reference 9
  4. lines 43-44
  5. lines 44-45
  6. corrected
  7. corrected
  8. lines 62-65

Materials and methods

  1. lines 70-71
  2. lines 73-74.
  3. corrected
  4. both groups are similar because they play in the same league in the Greek championship
  5. lines 88-90
  6. lines 90-93
  7. corrected.
  8. corrected

Results

  1. corrected
  2. corrected
  3. corrected
  4. corrected

Discussion

  1. corrected
  2. corrected 
  3. corrected 
  4. corrected

Conclusion

  1. corrected
  2. corrected
  3. corrected

Round 2

Reviewer 2 Report

Comments and Suggestions for Authors

1.      Although the authors have revised the manuscript, several points have not been improved. For example, in the introduction, there are areas for improvement, such as defining specialized terms and engaging with contrasting views. This could strengthen the introduction without adding more explanation or adding more details about the contractile alterations linked to neuromuscular control impairments related to concussions, especially the muscles around the knee joint in footballers.

Author Response

Hello...i will make the intro more attractive.Thanks for your review